# *Styrax* spp.: Habitat, Phenology, Phytochemicals, Biological Activity and Applications

**DOI:** 10.3390/plants14050746

**Published:** 2025-03-01

**Authors:** Antonello Paparella, Annalisa Serio, Liora Shaltiel-Harpaz, Bharadwaj Revuru, Prasada Rao Kongala, Mwafaq Ibdah

**Affiliations:** 1Department of Bioscience and Technology for Food, Agriculture and Environment, University of Teramo, Via R. Balzarini 1, 64100 Teramo, Italy; apaparella@unite.it (A.P.); aserio@unite.it (A.S.); 2Migal Galilee Research Institute, P.O. Box 831, Kiryat Shmona 11016, Israel; liora@migal.org.il; 3Environmental Sciences Department, Faculty of Sciences and Technology, Tel Hai College, Kiryat Shmona 11016, Israel; 4Newe Yaar Research Center, Agricultural Research Organization, P.O. Box 1021, Ramat Yishay 30095, Israel; bharat.revuru@gmail.com (B.R.); prasada.kongala@shiats.edu.in (P.R.K.)

**Keywords:** economic importance, integrative medicine, phytochemicals, Styrax

## Abstract

Styrax is the largest genus of the family Styracaceae, with about 130 species distributed across America, Europe, and Southeast Asia. The oleo-resin of these woody shrubs, called Styrax benzoin, has a long tradition of use as incense and in therapeutics, which has stimulated research and industrial applications. Many studies have been carried out on the biological applications of different Styrax species, but some gaps still remain to be filled, particularly regarding the phenology and the biological activity and application in different fields. Hence, this review gathers updated and valuable information on the distribution and phenology of *Styrax* spp., considering their phytochemicals, biological activity, current and possible applications in medicine, animal feeding, energy production, and the food industry. Overall, *Styrax obassia* and *Styrax japonicus* are the most studied, but *Styrax officinalis* has been thoroughly investigated for its phytochemicals. The recent literature highlights promising applications in oncology and also as an energy crop. The data described in this review could be useful in upgrading the quantity and quality of Styrax benzoin, as well as expanding knowledge on emerging applications, such as bio-pesticides or the development of active packaging for the food industry.

## 1. Introduction

The family Styracaceae of the order Ericales consists of 11 genera and about 160 species of dicotyledonous woody shrubs. The family is distributed across the regions of North America, South America, Europe, and Southeast Asia. In the American continent, the species spread from North America to Argentina, whereas in Asia, they spread from Japan to Eastern India, and one species occurs in the Mediterranean region. However, other genera of Styracaceae are confined to Eastern Asia. Paleo botany suggests that the family originated in the Eurasian region and was distributed across the world; fossils of various genera were obtained from the Americas, France, the Eurasian continent, and Japan. The 12 genera of the family include Styrax, Huodendron, Bruinsmia, Alniphyllum, Parastyrax, Halesia, Melliodendron, Rehderodendron, Perkinsiodendron, Changiostyrax, Pterostyrax, and Sinojackia. During the earlier notions of traditional taxonomy, Styracaceae was placed under various families of the order Ebenales, later classified into a separate family. With the evidence from molecular and morphological data, Styracaceae is known to be monophyletic in origin, with Styrax and Huodendron in one clade and the rest in another. The members of the family can grow into large trees 20–30 m tall, and a few species occur as shrubs (1–4 m) [1,2,3].

The leaves are simple, alternate, petiolated, pinnately nerved, stipulated with stellate or peltate trichomes. The inflorescence is terminal with axillary cymes, racemes, or panicles with 1–2 bracteolate flowers. The flowers are bisexual and actinomorphic, with hypanthium adnate to the ovary walls. The calyx is synsepalous, and the corolla is sympetalous. The stamens are uniseriate, usually 2 to 4 in number, equal to lobes of the corolla; the ovary is carpellate, septate at the base, partly to completely inferior, variable according to genera, with a filiform style and a terminal stigma. The ovules are unitegmic/bitegmic, tenuinucellate, anatropous, 4–9 in number/carpel, with a parenchymatous mesotesta, a seed coat without vascular bundles, a multiplicative and sclerotic outer mesotesta, and vascular bundles in the testa. Pollens are fairly uniform, solitary, radially symmetrical, tricolporate with a tectate–perforate–columellate exine. Fruits are dry with capsular, dehiscence, indehiscent, or drupaceous with persistent calyx. Seeds are globose to fusiform, rarely winged, and the endosperm is copious and oily. Further, the chromosomal number varies within genera, where Styrax has 8 chromosomes, while in Halesia, Sinojackia, and Pterostyrax, the chromosome number is 12 [1].

The genus Styrax is the largest of the family Styracaceae with about 130 individual species, which represent 80% of the family species. The name Styrax is an ancient Greek word meaning “fragrant resin”, which is predominantly produced in these species [2,3,4]. The genus is widely spread across America, East Asia, the Middle East, and the Mediterranean coast. These species grow in a wide range of climates depending on the region, including humid, warm temperate, lowland, and mountain ranging from an altitude of 100 m to 3000 m. A few species are xerophytic and native to the Mediterranean and grow in dry climates, and the plant morphology is modified accordingly to withstand harsh conditions. Molecular and evolutionary analyses suggest that the Styrax genus originated from Eurasia, followed by initial dispersal to Southern America, and a few other reports suggest that the species native to the subtropics originated from South America, followed by distribution throughout the current range. On the other hand, Styrax has only three species with entire leaf margins distributed in western Eurasia and western North America. Further, fossil records suggest that Styrax once had a wide distribution from the Northern Hemisphere to the Eurasian region [2,3,4,5].

The aim of this review is to provide a comprehensive and updated analysis of the scientific literature regarding the Styrax genus, including its distribution, classification, phenology, secondary metabolites accumulated in different tissues, and their antioxidant, antimicrobial activities, nutraceutical potential, medical, and industrial applications, with particular emphasis on the Mediterranean species *Styrax officinalis*, known for its high-value resin. Studies on *Styrax* spp. published before January 2025 were collected from scientific databases (Google Scholar and Web of Science, ResearchGate, PubMed, and publishers’ websites), using the keywords “Styrax”, “resin”, “benzoin”, “biological activity”, “uses”, “properties”, “applications”, and combinations thereof.

## 2. Distribution

The Styrax genus morphologically differs from the other genera by characteristics like a perfect flower with an inflorescence of racemes, cymes, or panicles, a superior ovary, and a drupaceous fruit [4]. This genus was initially classified into three sections [6], namely Pamphillia (South American species), Foveolaria, and Styrax. Later, Styrax was reclassified into two sections based on morphology: Valvatae and Styrax [5]. Valvatae is a predominantly tropical evergreen forest species distributed in Asia and Southern America, and it is subdivided into two series: Valvatae and Benzoin. Valvatae is a neotropical clade with 77 species containing wet mesocarp and endocarp, with ellipsoid seeds, whereas Benzoin, with nine species, is a paleo tropic having dry mesocarp and endocarp with globose seeds. The sections of Pamphillia and Foveolaria of Perkins classification were merged into Valvatae by Fristch. Initially, Pamphillia was considered an independent genus of Styracaceae, but then it was merged with Styrax due to the similarities in floral and vegetative parts and cooccurrence of gyno-dioecy, while the Styrax section was again subdivided into Cyrta and Styrax. The Cyrta series has 33 species with denticulate/serrated leaf margins and axillary/terminal inflorescence. Cyrta series accounts for 90% of species in the Styrax section, which are native to eastern North America and East Asia. Styrax series has only three species, two in the United States and one in the Mediterranean, with entire margins and terminal inflorescences [7].

The members of Styrax occur as shrubs or trees based on the geographical region and have an average 180-day life cycle. Due to their widespread distribution, species of Styrax genus display morphological variations to adapt themselves to the environment. For example, Brazilian Styrax species of two regions, Cerrado (*S. camporum*; *S. ferrugineus*) and Atlantic forest (*S. latifolium*, *S. martii*, and *S. leprosus*), differ in their wood morphology such as length, diameter, vessels, and fiber length. Further, it has been reported that Cerrado species have longer root vessels, whereas the Atlantic ones have longer stem vessels [8]. In Southeast Asia, *Styrax* spp. is endangered by climate changes, natural disasters, and land conversion; it has been estimated that the probability of extinction of *S. sumatrana* in North Sumatra might increase in 2050 unless suitable conservation strategies are implemented [9] (Figure 1).

The morphological variations led to taxonomical disputes in the several species, which were reclassified several times. Huang et al. [10] reclassified 17 Styrax species of *Cyrta* series in the Asian region, spread from the west of Japan to Nepal and South of Japan to Sumatra; these species also display changes in morphological characteristics according to their habitat. Later, Li et al. [11] reclassified 11 species based on valvate aestivation of corolla lobes in Asia, whereas these species are spread from Eastern India to Southern China and Malesia, and grow in different habitats either as shrubs or trees. Styrax species of America vary in size: *S. grandiflora* often grows as a shrub with a flabellate branching system and rounded crown, *S. argenteus* of southern Mexico grows as a large tree that can reach 35 feet, while the largest is *S. glaber* of the Caribbean with the height of 40 feet [12]. Apart from morphology, Styrax species also vary in the ploidy levels, as a genome analysis of 15 species revealed ploidy variations ranging from diploid to hexaploid, while *S. hookeri* is an octaploid species [13].

## 3. Monophyletic Origin, Genetic Diversity and Polymorphism

Previous reports suggest that the orders Ericales and Ebenales are of a monophyletic origin, and the Styracaceae form a close relationship with Diapensiaceae based on morphological and molecular evidence [14,15]. Interestingly, the plastome analysis of Styracaceae revealed a similar structure of the plastidial genome with the exceptions of Alniphyllum and Bruinsmia, which have an inverted region of 20 kb [16]. Also, plastome analysis assisted in identifying the Styracaceae and its genera as monophyletic except for Halesia and Pterostyrax [16,17]. Yan and coworkers [16] analyzed the plastome sequences of Styracaceae and determined Parastyrax as a sister species to the other seven genera of the family with monophyletic origin; Styrax, Huodendron, and Alniphyllum-Bruinsmia are closely associated and form a sister clade within the family, whereas Halesia and Pterostyrax are of polyphyletic origin. Further, the molecular analysis of the Styrax genus using ITS region (nuclear) and *trnK*, *rpoC1*, and *rpoC2* regions (chloroplast) led to the identification of monophyletic origin among the four species of the genus; however, it failed to identify relations within the species. The disjunctive distribution of the genus was explained by the North Eurasian origin of the genus and later its migration to South America and East Asia [4].

Genetic variations are predominantly observed between the wild type and commercial cultivars of various Styrax species. Since Styrax is grown for the benzoin, these genetic variations might impact the amount and secondary metabolite contents of benzoin resins extracted from bark tissues, thereby influencing its trade value in global markets. For instance, the phylogenetic analysis of the Asian species *S. sumatrana*, *S. suberifolius*, and *S. chinensis* revealed their closest relationship; these species are supposedly native to China [18]. In addition, *S. sumatrana* exhibits intraspecific differentiation; molecular analysis using the *trnL-trnF* chloroplast region from the three populations showed a haploid nature and strong genetic differentiation, which occurred due to geographic barriers that affected free gene flow [19]. *S. japonicus* is a prominent member of the genus, having ornamental and medicinal properties. *S. japonicus* is a polymorphic species, and the commercial cultivars vary genetically. Chloroplast genome analysis of four *S. japonicus* subspecies/cultivars displayed the conserved nature of the genome followed by polymorphisms in the few sets of coding and noncoding genes that can be used as DNA barcodes for cultivar identification [20]. Positive selection pressure on two chloroplast genes (*ycf1*, *ndhD*) in analyzed samples of *S. japonicus* indicated the adaptive evolution of the species toward novel niches [20]. In addition, 23 microsatellite regions were identified on the genomes of *S. japonicus* and *S. confusus*, which are highly polymorphic and variable in nature [21]. Transcriptome analysis and identification of expression sequence tags (ESTs) of *S. japonicus* also indicated the polymorphic nature of the species [11]. Plastome analysis identified that *S. japonicus* is closely related to *S. grandifloras*, *S. confusus*, and *S. calvescens* [22]. Moreover, *S. obassia* and *S. magnus* also show high genetic diversity in the wild populations when compared to commercial cultivars [23,24]. Interestingly, *S. redivivus* displays genetic polymorphism and morphological changes within the species depending on the latitude changes. Accordingly, two different groups were suggested based on the morphology, but the data were not enough to delimit the taxa [25].

The prominence of genetic diversity within Styrax species led to several discrepancies in species identification and cultivation for commercial purposes. To authenticate and understand the phylogeny, scientists elucidated the chloroplast genomes of several East Asian species for identifying potential barcodes. Species such as *S. obassia*, *S. dasyanthus*, *S. zhejiangensis*, *S. macrocarpus*, *S. odoratissimus*, *S. calvescens*, and *S. japonicus* were analyzed for their chloroplast genome; it was observed that these species are related to each other and share a common ancestral origin [22,26,27,28,29,30]. In addition, after analyzing 36 chloroplast genomes of Styrax species, *ycf1b* and *trnT-trnL* regions were identified as potential barcode candidates to discriminate between the species [31]. Further, Susilowati and coworkers [18,19] used the combination of *matK* and *rbcL* genes to authenticate the Styrax species in the Sumatran region. Nevertheless, the attempts were not successful [18,19]. Recently, Feng et al. [32] used the ITS barcode to determine the authenticity of benzoin resin extracted from species and found that most of the analyzed samples were identified as *S. japonicus* and *S. tonkinensis* [32]. These results suggest that despite Styrax monophyletic origin, genetic variations occur within the species depending on the biogeography that affects the cultivation practices. In-depth analysis with the assistance of advanced omics techniques to understand these genetic variations and polymorphisms could lead to the development of superior varieties for escalated cultivation.

## 4. Economic Importance

Styrax species are predominantly grown for their resin and also as ornamental plants. The flowers are known for their fragrance, and the color ranges from white to pink, 1–2 cm in diameter, and blossoms during the mid-spring season. *Styrax japonicus* and *Styrax obassia* species are generally used for landscaping and are known for their white bell-shaped flowers, commonly called “snow bells”. Apart from their ornamental purposes, few species are cultivated for wood, resin, and seed oil [22].

Resins produced by Styrax are generally termed benzoin, and the composition of benzoin resin varies according to the species. Recently, He et al. [33] have overviewed the methods used for extracting benzoin resin and controlling its quality. Benzoins are generally categorized into two classes based on the content of cinnamic acid. Benzoin resin that contains this acid is called Sumatran benzoin, which is produced by Siam Styrax benzoin found on the islands of Java and Sumatra, while benzoin resin lacking cinnamic acid is called Siam benzoin or benzoe tonkinensis, produced by *S. tonkinensis*, native to the Indo-China belt. Free cinnamic acid, benzoic acid, benzyl cinnamate, coniferyl cinnamate, and sumaresinolic acid are the key components of Sumatra benzoin, whereas Siam benzoin has coniferyl benzoate, free benzoic acid, cinnamyl cinnamate, siaresinolic acid, sumaresinolic acid, and 3-oxosiaresinolic acid [34,35]. Analysis of different Siam benzoin and Sumatra benzoin resins by gas chromatography (GC-MS) and high-performance liquid chromatography (HPLC-MS) coupled with mass spectrometry revealed the presence of various secondary metabolites: benzoic acid, vanillin, isovanillin, benzyl benzoate, lubanol, benzyl cinnamate, *p*-coumaryl benzoate, coniferyl benzoate, *p*-coumaryl cinnamate, lupeol, oleanonic acid, cinnamic acid, ethyl cinnamate, and coniferyl benzoate [36] (Figure 2).

Benzoin resin is a pathological exudation produced from the injured stems after the injury through the secretory ducts. To collect the resin, farmers usually make an incision, followed by the exudation of transparent gum from the wound along with water and other substances; then, the gum becomes hard through evaporation and finally forms benzoin [33]. To upgrade the value of Styrax resin, Kholibrina and Aswandi [37] have proposed to process resin into an essential oil form, thus also obtaining a valuable hydrolate as a by-product when steam distillation is used.

Benzoin resin is widely used in medicinal and aromatic industries and possesses a significant market value [38]. Resins produced from *S. benzoin* are used to treat skin diseases, arthritis, wounds, muscle pain, and neurological disorders [39].

## 5. Antioxidant and Antimicrobial Activities of *Styrax* spp.

The use of *Styrax* spp. in traditional medicine often relies on their biological activity in terms of antioxidant and antimicrobial properties. Aqueous and organic solvent-based extracts of *S. officinalis* leaves showed potential antioxidant activity with IC_50_ values of 11.7 ± 0.53 and 15.5 ± 0.69 μg/mL, respectively. When S. *officinalis* was compared with Trolox (an IC_50_ value of 2.8 μg/mL), *S. officinalis* exhibited higher activity. Similar experiments also showed that the organic extract of leaves from *S. officinalis* emerged as the best antimicrobial agent [40].

Egonol, extracted from *S. officinalis* seeds, was demonstrated to exert antioxidant activity when applied to erythrocytes. In detail, superoxide dismutase (SOD) and glutathione-S-transferase (GST) levels were increased in erythrocytes exposed to increasing concentrations of egonol (0.2–1.0 mg/mL) for 15 min at 37 °C, while the concentration of other enzymes was not affected. The final effect was an accelerated detoxification of superoxide radicals to hydrogen peroxide, and the boost of the antioxidant defense system of erythrocytes [41]. Antioxidant activity, with no antibacterial effect, has been reported recently for the methanolic extract of *S. officinalis* fruit pericarp [42]. On the contrary, only a weak antioxidant activity was observed for the crude methanolic extracts of *Styrax camporum* and *S. ferrugineus* fruits (18.47 ± 1.54 and 2.40 ± 2.01 diphenyl-picro-hydrazyl (DPPH) % inhibition) with respect to gallic acid (92.41 ± 0.34 DPPH % inhibition), as reported by de Almeida Silva et al. [43]. Moreover, also the kemenyan resin obtained from *S. sumatrana* demonstrated radical scavenging and antioxidant activity according to DPPH assay, with IC_50_ values ranging from 15.28 to 31.74 mg/L [44]. Finally, egonol exhibits luminophore properties and emits in the near UV region along with electron-quenching properties [45]

Several researchers identified the beneficial properties of *Styrax* spp. extracts (extracted with organic solvents), and they are evident in the available scientific literature. The main findings regarding the antimicrobial activities of *Styrax* spp. and the derived molecules against pathogenic bacteria and fungi are reported in Table 1 and Table 2**,** respectively.

Aqueous and organic extracts of *S. officinalis* showed antioxidant activity as well as inhibitory effects against potential human pathogens such as *Staphylococcus aureus*, *Shigella sonnei*, *Escherichia coli*, *Pseudomonas aeruginosa*, *Enterococcus faecium*, *Candida albicans*, and *Epidermophyton floccosum* with the minimum inhibitory values ranging from 12.5 to 25 mg/mL [46].

Also, egonol and a few derived novel compounds from the stem of *S. officinalis* showed antibacterial activity. Interestingly, *Styrax* spp. extracts display potential antifungal activity against various pathogens, although the effect varies depending on the species. For example, the *S. officinalis* extract determined the total inhibition of Phytophthora *infestans,* and it reduced the growth of *Alternaria solani* by 50.58% [50,53].

Essential oils produced from resin are used in food, as flavoring agents in beverages, and as wood varnish. They have mind-relaxing properties [54] and could also be used for their antibacterial activity. Silver nanoparticles produced from resin essential oil possess antimicrobial activity against Gram-negative and Gram-positive pathogens and display radical scavenging activities. Molecular docking studies revealed the binding of essential components to the pathogenic bacterial enzymes and the reduction in their activity [55].

Further, various bacterial and fungal endophytic microorganisms were isolated from the tissues of *S. officinalis*, and the cell-free extracts of these endophytes were demonstrated to possess antimicrobial activity against human pathogens [56].

## 6. Applications of *Styrax* spp. in Integrative Medicine

Integrative medicine is a promising branch of evidence-based medicine that combines conventional, complementary, and alternative medicine [57]. In this context, herbal plants and, most of all, medicinal plants have a long tradition of bringing medicine back to its roots and stimulating the interest of both researchers and the population. Different *Styrax* species have been investigated for their pharmacological potential, with *S. obassia* and *S. japonicus* emerging as the most extensively studied species [58]. Moreover, Chinese researchers carried out studies using the so-called “false Styrax”, also named Storax (Liquidambar orientalis), which belongs to a different genus and, therefore, will not be considered in this review [59].

The pharmacological properties of various *Styrax* spp. and the uses in traditional medicine encompass a wide range of biological activities: antimicrobial, antioxidant, antiinflammatory, antiproliferative, cytotoxic, anticomplement, promoting the biosynthesis of estrogen, inducing apoptosis, acetylcholinesterase inhibitor, antiasthmatic, antiulcer, inhibitory effect for interleukin, and matrix metalloproteinases activity [58].

In the last decade, the scientific literature on the pharmacological properties of *Styrax* spp. has focused on possible applications in oncology, although previous studies already demonstrated interesting antiproliferative effects of triterpenes extracted from *S. tonkinensis*, especially olenoic acid, on human promyelocytic leukemia cell line HL-60 [60]. Recently, a novel hybrid dioxime ligand has been developed using egonol from *S. officinalis*, and the ligand has been applied to cancer cell lines to determine its efficacy. The ligands inhibited cell migration and induced apoptosis in the cell lines at 5–40 µM concentration [51]. Moreover, egonol glucoside obtained from the bark of *S. obassia* showed in vitro cytotoxic activity against the breast cancer cell line Michigan Cancer Foundation (MCF)-7, the human cervical cancer cell line Hela, and the HL-60 cell line; in addition, castanoside B from the same raw material demonstrated cytotoxic activity against Hela and MCF-7 cells [61]. Similar results were observed with egonol obtained from the purification of methanol extracts of *S. camporum* and *S. ferrugineus* [43]. Finally, methanolic extracts of *S. officinalis* leaves were observed to be cytotoxic against human hepatocellular liver carcinoma and mammary gland carcinoma cell lines with IC_50_ values between 14.95 µg/mL and 19.38 µg/mL [62].

## 7. Styrax Benzoin as an Additive for Foods and Feedstuffs

The balsamic resin of *S. tonkinensis* is a food and feedstuff flavoring agent. The chemical and technical assessment of the safety of this additive was initiated in 2010 within the FAO/WHO Expert Committee on food additives and ended in 2014 with the safety evaluation (FAO and WHO, 2015). Given the NOAL (No Adverse Effect Level) of 500 mg/kg bw per day, the committee concluded that the resin would not be of safety concern when used as a flavoring agent.

In the European Union, the same resin is listed among the flavoring agents approved as feed additives, in the functional group 2b (aromatizers) included in Annex I of Regulation 1831/03/EC, although authorization expired in 2023.

In the United States, benzoin resin from *Styrax* spp. is approved by the Food and Drug Association (FDA) as a flavor and flavor enhancer, with the CAS (Chemical Abstract Service) No. 9000-05-09. In China, the resin obtained from *Liquidambar* spp. is approved as a food flavor with the FEMA (Flavor and Extract Manufacturer Association) No. 3036. *Liquidambar* spp. extracts or distillates are also regulated as a fragrance ingredient with a specific standard issued by the International Fragrance Association (IFRA); according to this standard, crude gums shall never be used as a fragrance ingredient for any finished product application, due to the risk of dermal sensitization (IFRA Amendment 49, available at https://www.perfumersworld.com/documents/Project-Templates/IFRAStandards/Styrax.pdf, accessed on 30 June 2021).

Recently, benzoin oleogum resin (BOGR) has been used to develop active films for food packaging. Bathia et al. [63] loaded antioxidant gelatin/pectin films and evaluated several technological and functional properties, finding that BOGR increased the antioxidant activity and the thermal stability of the packaging material, although water permeability, opacity, and thickness were increased.

## 8. *Styrax officinalis*

*Styrax officinalis* L. (Styracaceae) is a perennial deciduous small tree or shrub, famous for herbal medicine [40,64].

Among traditional medicinal plants, *S. officinalis* L. is considered the prescribed source of herbal medicine for antiseptic and antirespiratory diseases. It is one of the oldest and most famous medicinal plants from ancient times. In the last two decades, the global revolution in the search for bioactive herbal products has led to a marked increase in both the scientific literature and patents, in particular for the treatment of chronic human diseases in the frame of integrative medicine. In fact, conventional drugs may fail for several reasons due to a narrow spectrum of activity that is used to lessen the symptoms, antimicrobial resistance, as well as limited efficacy, e.g., for the treatment of cancer, metabolic disorders, autoimmune diseases, viral infections, and many others [40].

*S. officinalis* L. is able to grow extensively at altitudes between 0 and 1000 m [64]. The plant is a deciduous shrub that grows up to 6 m in height. The leaves are simple, alternate, soft, rounded, or oval-shaped and have a greenish-white color with hairy lower surfaces. The flower is white in color and bell-shaped. The ovary of the flower is usually longer with one pattern, 8–16 stamens, and twice the number of petals. The fruits are hairy ball-shaped drupes, greenish-yellow with lenient and lengthy apexes [46]. Dried fruits of *S. officinalis* are of different sizes ranging from 10 to 25 mm and occur in round/spherical shape, smooth and shiny; the color ranges from light brown to grayish and shows a distinct scar (hilum). Dry fruit is tough with one or two, sometimes three large ovoid seeds. The tissue pericarp is different, with three distinct layers: a light brown outer layer, called the exocarp; the inner layer, called the endocarp, with a hard shell surrounding the seed; and a bright, yellowish transparent mesocarp [65]. *S. officinalis* is the only species identified as a drought-tolerant plant among the members of the *Styrax* genus. Tayoub et al. [66] observed the presence of stellate hairs on the aerial parts and secretory glands in the leaf parenchyma as an adaptive strategy to survive drought conditions. Plants usually germinate from seeds by mechanical cracking of seeds, and germination occurs after 21 days during the months between May and September [67]. The species generally produce a typical benzoin resin called solid storaque, which is widely used in perfumery [34] (Figure 3).

*S. officinalis* L. is considered native to arid regions of the Mediterranean Basin and Arabian Plateau and is also found in Central America, Mexico, Southeastern Europe, Southwestern Asia, and the Mediterranean region, including West and South Anatolia [4,12,40]. The focal region of the spreading of *S. officinalis* L. links with the coasts of the eastern Mediterranean and the islands between eastern Greece and Israel, mainly Crete and Cyprus. In Turkey, it is intensively distributed beside the west and south coasts. However, the species infiltrates deeper into central Turkey [68]. The presence of this plant has been known since ancient times. The ancient Greek geographer Strabo, while describing the city of Selge in his book “Geographika”, mentioned the rich vegetation of the region as well as the Styrax tree [64] (Figure 4).

## 9. Phytochemicals of *S. officinalis*

### 9.1. Lignans

The lignan constituents, isolated from the nucleus of *S. officinalis* stem, are benzofuran, tetrahydrofuran, and furofuran lignans [69]. A naturally occurring benzofuran glycoside, namely 5-(3-hydroxypropyl)-7-methoxy-2-(3,4-methylenedioxyphenyl) benzofuran, known as egonol, is widely present in *S. officinalis* [40,51]. Pazar et al. [70] identified five benzofurans from the dried endocarp of *S. officinalis*: americanin A, egonololeat, egonol-2-metilbutanoat, egonolgentiobiside, and homoegonolgentiobiside—out of which americanin and egonololeat were reported for the first time in the *Styrax* genus. Similarly, several lignans and triterpenes were isolated for the first time from the leaves of *S. tonkinensis* [71]. Akgul et al. [72] isolated several benzofurans that are derivatives of egonol from the seeds and the stem extracts of *S. officinalis*: egonol-*β*-gentiotrioside, 2-(3,4-dimethoxyphenyl)-5-(3-hydroxypropyl)-7-methoxy-benzofuran-β-gentiobioside,egonol-*β*-gentiobioside, egonol-*β*-glucoside, and 5-[3″-(2-methylbutanoyloxy) propyl]-7-methoxy-2-(3′,4′-dimethoxyphenyl) benzofuran (Figure 5).

Hexane extracts of the *S. officinalis* seeds revealed the presence of two new benzofurans, (i) 5-(3″benzoyl oxy propyl)-7-methoxy-2-(3′,4′-methylenedioxyphenyl)-benzofuran and (ii) 4-[3″-(1c-methylbutanoyl oxy) propyl]-2-methoxy-(3′,4′-methylenedioxyphenyl)-1a, 5b-dihydrobenzo-[3,4]-cyclobuta oxirene, along with four other benzofurans: (i) 5-[3″-(1c-methylbutanoyloxy) propyl]-7-methoxy-2-(3′,4′-dimeth oxy phenyl)-benzofuran; (ii) 5-[3″-(1c-methylbutanoyloxy) propyl]-7-methoxy-2-(3′,4′-methyl enedioxyphenyl)-benzofuran; (iii) 5-(3″-acetoxypropyl)-7-methoxy2-(3′,4′-methylenediox phenyl)-benzofuran; and (iv) 5-(3″-hydroxypropyl)-7-methoxy-2-(3′,4′-met hylenedioxy phenyl)-benzofuran. These four benzofurans were also reported from the seeds of *S. obassia* [72]. Similarly, five benzofurans, namely (i) erythrodiol-3-acetate, (ii) stigmasterol, (iii) (24R)-24-ethylcholesta-4,22-dien-3-one, (iv) 5-(3″-hydroxy propyl)-7-methoxy-2-(3′,4′-methylenedioxyphenyl) benzofuran, and (v) 5-(3″-hydroxy propyl)-7-methoxy-2-(3′,4′-dimethoxyphenyl) benzofuran, were isolated from the bark of *S. benzoides*; chemotaxonomy suggests that *S. benzoides* is closely associated with *S. officinalis* [73]. On the other hand, four benzofurans, namely (i) egonol gentiobioside, (ii) egonol gentiotrioside, (iii) egonol glucoside, and (iv) masutakeside, were isolated for the first time from the fruits and bark of *S. benzoin*, with structures that were quite similar to the benzofurans of *S. officinalis* [74]. Courel et al. [75] identified egonol, homoegonol, and styraxin as significant lignans in the balsams of *S. officinalis*.

### 9.2. Terpenoids

A pentacyclic triterpenoid, and later, three other triterpenoid saponins, Styrax-saponins A–C, and a deacyl saponin were isolated from the extracts of *S. officinalis* L. Further, it has been reported that *S. officinalis*, native to Syria, is rich in triterpenes such as saponins and sapogenins [76]. Tayoub et al. [66] extracted essential oil (EO) from the leaf, flower, and stem tissues of *S. officinalis* and reported the content of terpenes. It has been identified that terpene content varied between 41 and 48% in all analyzed tissues. Besides terpenes, lipids, aromatic compounds, and hydrocarbons were predominant in the EO. Specifically, *S. officinalis* leaf EO contained terpenes like hexenal, linalool, and geranial, while the stem EO contained significant amounts of *α*-terpineol, eugenol, and benzyl benzoate. Further, the contents of linalool, dodecane, and tridecanal were higher in the flower EO [66]. In continuation of the previous study, the authors of ref. [77] evaluated the phenological changes in the EO of *S. officinalis* native to France. They observed the production of EO, which was yellowish with a fresh odor and had a different composition compared to the other phenological stages of the plant; lipid derivatives and monoterpenes were significantly higher in the EO, while sesquiterpenes were higher in the leaves of the vegetative stage compared to other stages. Three components, namely (*E*)-2-hexenal, octanol, and geraniol, were significantly higher in all the EOs. On the other hand, terpinen-4-ol, *p*-cymen-8-ol, geranial, and linalool were higher at the flowering stage, while (*E*,*E*)-2,4 heptadiene, (*E*,*E*)-2,4 decadienal, and geranylacetone were higher during the vegetative stage, and only piperanal content was significant during the fruiting stage [77]. A comparative study on the floral emissions of three Styrax species (*S. japonicus*, *S. grandiflora,* and *S. calvescens*) resulted in the identification of various volatile terpenes, predominated by *α*-pinene, myrcene, linalool, germacrene D, ocimene, allo-ocimene, and *β*-elemene; a few of these compounds were also reported from the flower EOs of *S. officinalis*, indicating a similar chemical profile [66,78] (Figure 6).

### 9.3. Lipids and Other Secondary Metabolites

Analysis of the fruit of *S. officinalis* revealed 15 compounds belonging to seven different classes of natural metabolites with chemotaxonomic relevance: tri-*α*-linolenoyl-sn-glycerol; 1,2-di-*α*-linolenoyl-3-linoleoyl-sn-glycerol; 1-*α*-linolenoyl-2-palmitoyl-sn-glycerol; 1,2-di-*α*-linolenoyl-sn-glycerol; egonol; demethylegonol; homoegonol; 1,5-anhydro-D-mannitol; glucose; sucrose; 6′′′-O-benzoyl-sucrose; raffinose; lactic acid; succinic acid; and glutamic acid. Furthermore, *S. officinalis* is a valuable source of enantiopure 1,5-anhydro-D-mannitol, which has several medicinal potentialities and is a useful building block in organic synthesis, in particular in “green” approaches of valuable and potentially biologically active molecules [79].

In plant tissue culture, treatments of stem calli with boron, niacin, and thiamine diphosphate induced the production of benzoin between 166 and 231%. Further, benzoin possesses various volatile organic compounds such as hexane, cyclohexane, and cyclopentane [80]. Seeds of *S. officinalis* were analyzed for lipid and protein content at different harvesting times, and seeds were rich in unsaturated fatty acids such as linoleic acid (66–73%) and oleic acid (16–22%). Interestingly, oil content increases by prolonging the harvest time; also, fatty acid contents vary between the first and second harvest. Linoleic acid content was higher during the second harvest, while oleic and palmitic acids were higher in the first harvest [81]. Further, ursolic acid, sumaresinolic acid, and the latter’s derivatives were identified as predominant triterpenes in the resins of *S. officinalis*, along with several phenolics and flavonoids, namely cinnamyl cinnamate, methoxy eugenol, cinnamic acid, benzyl cinnamate, cinnamaldehyde, vanillin, eugenol, vanillic acid, benzyl benzoate, cinnamyl benzoate, ferulic acid, cinnamyl alcohol, propyl cinnamate, coniferyl alcohol, and 3-phenylpropanyl cinnamate [36]. Dib et al. [82] analyzed the ethanolic extracts of the fruit of *S. officinalis*, and found that the pericarp contained several active phytochemicals, such as tannins, saponins, and triterpenes, whereas alkaloids and flavonoids were absent. These extracts did not show any antimicrobial activity but displayed strong ichthyotoxic and molluscicidal effects. The toxicity tests with the snail *Cornu aspersum* showed a potent molluscicide activity, and the contact of snails with the extracts led to severe dehydration and foaming, ultimately leading to death in 30 min. However, ingestion of the extract did not show any toxic effects on snails

## 10. Specific Applications of *Styrax officinalis*

Since ancient times, *S. officinalis* L. has been famous for the preparation of perfumes and incense, as well as for its pharmaceutical properties [68,83]. *S. officinalis* was intensively used in traditional medicine for the treatment of several diseases until the 19th century when the pharmaceutical industry started to produce officially approved drugs [40]. Crude extracts and fractions obtained from different plant parts exhibited several medicinal properties, viz., antioxidant, antimicrobial [84], antihemorrhoid [85], and antifungal [50] activities.

Phytochemical analysis of the whole plant of *S. officinalis* resulted in the identification of cardiac glycosides, saponin glycosides, alkaloids, phenols, volatile oils, tannins, steroids, and flavonoids. Moreover, dry fruit and seed coat extracts of *S. officinalis* have allelochemicals that can inhibit seedling germination. The water extracts of *S. officinalis* significantly inhibited *Salvia sclarea* seedling growth compared to tap water controls. The allelopathic effect of *S. officinalis* was confirmed in another study, where scientists treated fennel, coriander, and fenugreek seeds with extracts of *S. officinalis* seeds, which inhibited the germination of all seeds [86].

## 11. Applications of *Styrax officinalis* as a Forage Crop or Energy Crop

*S. officinalis* leaves of different phenological stages have been tested as forage for ruminants. Esen et al. [87] evaluated the nutritional value of this species for goat feeding and found that metabolizable energy and degradability of dry matter and crude protein ranged from moderate to high; although affected by phenological stages, the nutritional value was interesting for application as forage until the end of the flowering stage. In this study, leaves were also used for in vitro gas production, yielding 20–27 mg/200 mg of dry matter. In another report, leaves collected at different periods (May, July, and September) were analyzed for organic matter digestibility and in vitro biogas production. It has been observed that values of crude ash, dry matter, crude fat, acid detergent fiber, and neutral detergent fiber were significantly higher in September compared with other months [88]. Also, a recent report suggests that forage of *S. officinalis*, for its high nutrient and fiber content, can be used as an effective food for livestock [89].

*S. officinalis* is considered an alternate oilseed plant in the form of perennial shrubs [64] as its seeds are rich in fatty acids and lipids that can be used for biodiesel production [90,91]. Palmitic acid, oleic acid, and linoleic acid are the primary composite of *Styrax* seed oil; fatty acid and protein analyses of five Styrax seed samples revealed the uniform fatty acid composition that could be used for developing biodiesel [90]. A study conducted on *S. tonkinensis*, which is also used as a silviculture plant to maintain forest lands and ecosystems, revealed that its oil content generally improves post 140 days of flowering, and the precursor pool for the fatty acid biosynthesis is obtained predominantly from the pyruvate, acetaldehyde, and phosphohexose pathways [92]. Both the seeds and foliage are widely used for biodiesel. Several scientific groups successfully produced biodiesel from the fresh biomass of *S. confusus* and *S. tonkinensis* through transesterification, and the yield was up to 65–70% [93,94].

Yesilyurt et al. [64] used *S. officinalis* L. as a novel material for synthesizing biodiesel, applying the Taguchi experimental design to identify the optimal reaction conditions. Significantly, an oil content of 48.29 ± 3.81% was obtained, with a theoretical maximum yield of 89.23% biodiesel at optimal conditions: concentration of catalyst 0.6 wt%, methanol/oil molar ratio of 6:1, duraton of reaction about 60 min, and the temperature of 60 °C. In this study, the most significant factors for biodiesel yield were catalyst concentration and methanol-to-oil molar ratio, with the contributing factors of 78.07% and 20.32%, respectively. The properties of the fuel—the methyl esters from *S. officinalis* L. oil—were within the ranges of the European Norm (EN) 14214 specifications [64].

## 12. Applications of *Styrax officinalis* as a Bio-Pesticide

Styrax species, particularly *Styrax officinalis*, have demonstrated potential as natural pesticides due to their bioactive compounds. Hepper [95] reported for the first time the use of a powder for poisoning sea fish off Lebanon. The powder is prepared from the seeds of storax (*S. officinalis* L.) and tubers of common cyclamen Cyclamen *persicum* Mill. Another study by Dib et al. [82] investigated the bioactivity of *S. officinalis* fruit extract. The extract exhibited strong ichthyotoxic and molluscicidal effects. In toxicity tests, snails (*Cornu aspersum*) exposed to surfaces treated with a 1% (*w*/*v*) pericarp extract experienced severe dehydration and foaming, leading to death within 30 min. However, ingestion of treated lettuce leaves did not produce observable effects on the snails. These findings suggest that *S. officinalis* fruit extract is a promising natural contact molluscicide with minimal impact when ingested by non-target organisms.

Additionally, a review by Xia [58] discusses the chemical constituents and biological activities of the genus Styrax. The review highlights that various Styrax species contain compounds such as lignans and terpenoids, which exhibit significant pharmacological activities, including cytotoxic, acetylcholinesterase inhibitory, antioxidant, and antifungal effects. These properties further support the potential use of Styrax species in developing natural pesticides.

## 13. Conclusions

The interest in *Styrax* spp., Styrax benzoin resin, and their applications is demonstrated by the growing number of papers that have been published in the last decade. Recent studies have focused on possible applications in integrative medicine, especially in oncology, as well as animal feeding or energy production. Emerging applications might include the development of novel active packaging for the food industry. These studies also underscore the potential of Styrax species as sources of natural pesticides, particularly as contact molluscicides, warranting further research into their efficacy and safety in pest control applications.

Although these plants can grow in the desert, they can be endangered by climate changes and, therefore, need preservation strategies, especially in Southeast Asia. In this paper, we have highlighted the significant scientific gaps in the research on the Styrax genus that need to be addressed to develop highly resilient and sustainable varieties for global cultivation practices. Throughput analysis of the genetic and allelic variants within the species/genera and identifying the key factors that regulate the production of resins could aid conventional/molecular breeders and biotechnologists in developing high-yielding varieties. Whole-genome sequencing combined with multi-omics of the economically important Styrax species will be highly helpful in species conservation across the wild and within laboratories. The information collected in this review can help find new solutions to save this valuable crop from extinction while upgrading the quantity and quality of benzoin resin, which is also a source of bioactive compounds with promising and emerging industrial and medical applications.

## Figures and Tables

**Figure 1 plants-14-00746-f001:**
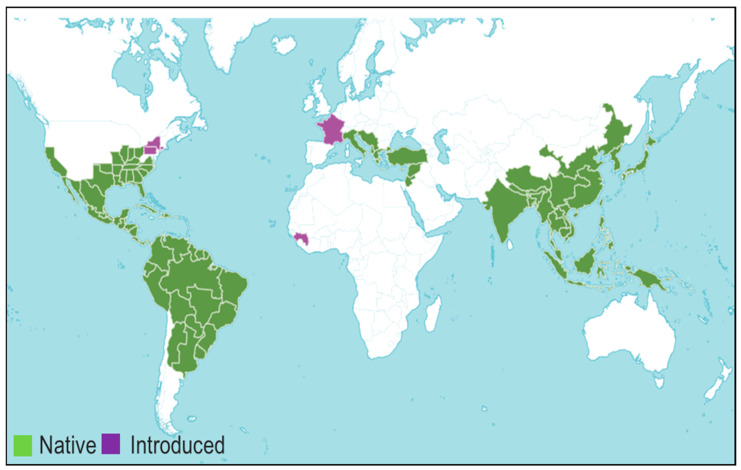
Geographic distribution of botanical species belonging to the genus Styrax (https://powo.science.kew.org/taxon/urn:lsid:ipni.org:names:327016-2#source-KBD, accessed on 21 May 2024).

**Figure 2 plants-14-00746-f002:**
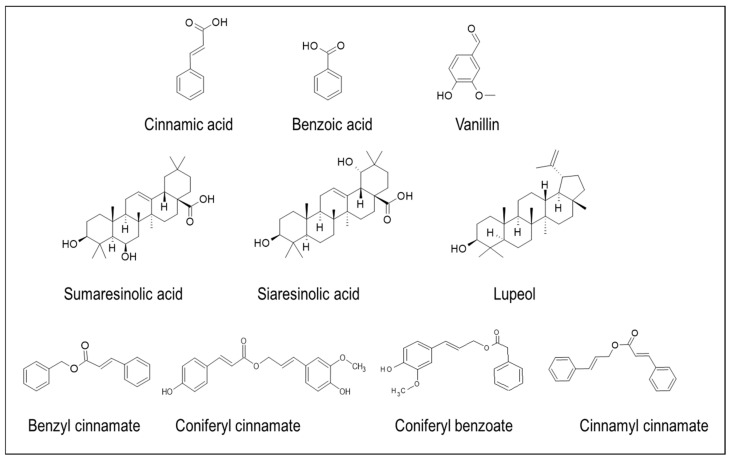
Major chemical compounds found in either Sumatra benzoin and/or Siam benzoin.

**Figure 3 plants-14-00746-f003:**
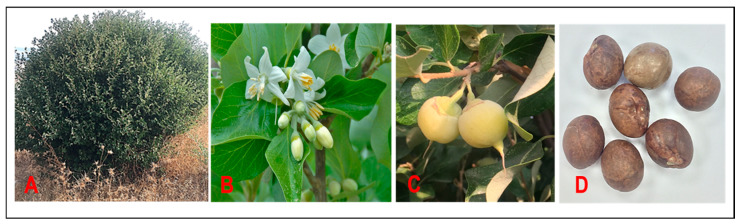
Photograph of *Styrax officinalis* L. of the Kabul Mountain (North Israel). (**A**) Styrax shrub/tree; (**B**) flowers; (**C**) fruits; (**D**) seeds (personal photos).

**Figure 4 plants-14-00746-f004:**
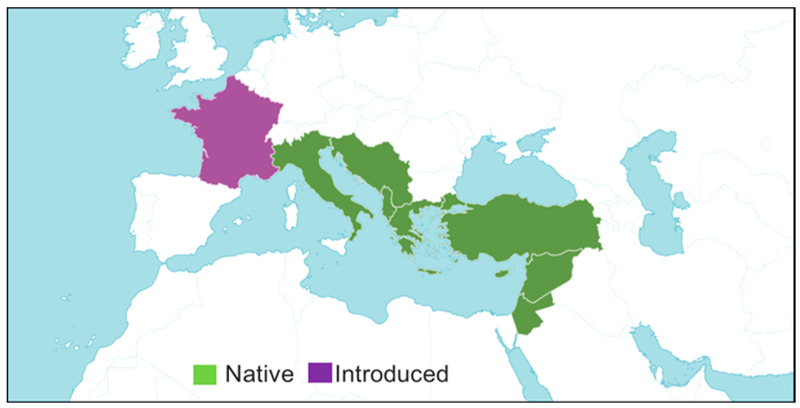
Distribution map of *Styrax officinalis*. *S. officinalis* distribution map in the temperate and subtropical regions of the Mediterranean Basin, Middle East, and various parts of Western Europe (https://powo.science.kew.org/taxon/urn:lsid:ipni.org:names:327016-2#source-KBD, accessed on 21 May 2024).

**Figure 5 plants-14-00746-f005:**
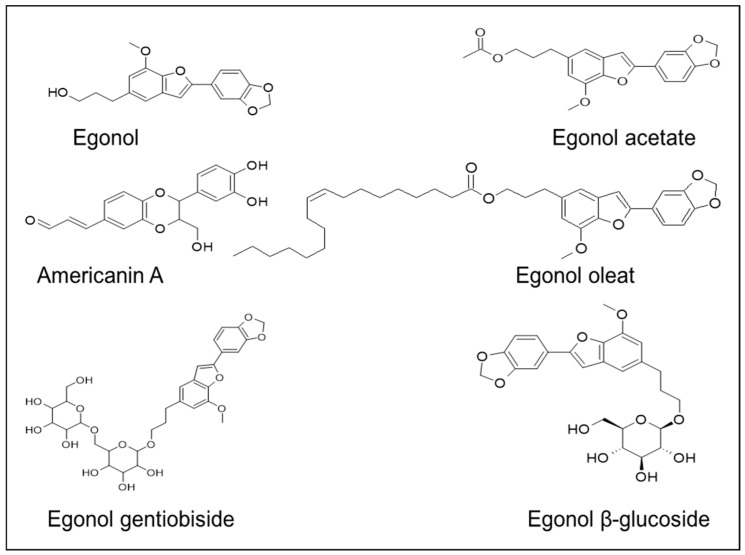
Lignan compounds isolated from the *S. officinalis* stem.

**Figure 6 plants-14-00746-f006:**
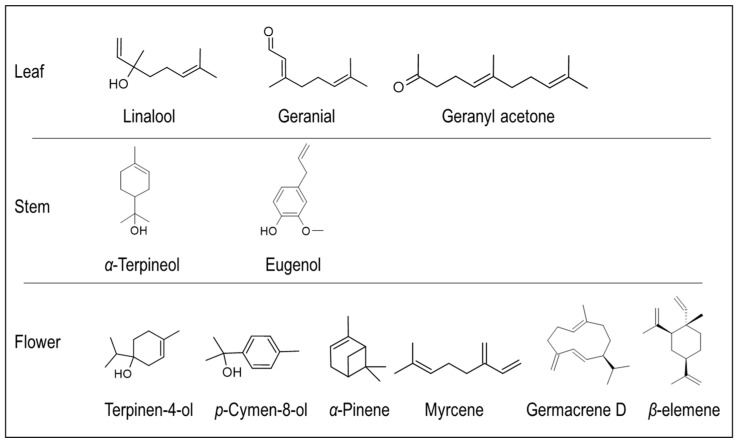
Major terpene volatile compounds isolated from the *S. officinalis* L. essential oils of different tissues.

**Table 1 plants-14-00746-t001:** Antibacterial activity of *Styrax* spp. and derived molecules.

Species/Compound	Part of the Plant	Analytical Assay	Minimal Inhibitory Concentration	Target Bacteria	Reference
*S. officinalis* organic extract	Aerial part	Minimum Inhibitory Concentration Assay	12.5 mg/mL 12.5 mg/mL 3.125 mg/mL 6.25 mg/mL	*Staphylococcus aureus*, *Enterococcus faecium*, *Methicillin-resistant**S. aureus* (MRSA) *Pseudomonas aeruginosa*	[46]
*Styrax pohlii* n-hexane fraction and ethyl acetate fraction	Aerial parts	Microdilution Method	200 μg/mL	*Streptococcus pneumoniae, Haemophilus influenzae* (EtOAc fraction)	[47]
Crude extract (CH_2_Cl_2_–CH_3_OH) of *S. ferrugineus*	Leaves	Thin Layer Chromatography (TLC) Plates	200 mg/mL	*Staphylococcus aureus*	[48]
Egonol, egonol glycoside (*S. ferrugineus)*	Leaves	TLC Plates	10–20 mg/mL	*Staphylococcus aureus*	[48]
Egonol (*S. officinalis*)	Fruit	Mueller Hinton Broth Method	800 mg/mL	*Staphylococcus aureus*, *Bacillus subtilis*, *Escherichia coli*	[49]
Homoegonol, homoegonol glucoside (*S. ferrugineus*)	Leaves	TLC Plates	10–20 mg/mL	*Staphylococcus aureus*	[48]

**Table 2 plants-14-00746-t002:** Antifungal activity of *Styrax* spp. and derived molecules.

Species/Compound	Part of the Plant	Analytical Assay	Minimal Inhibitory Concentration	Target Fungi	Reference
*S. officinalis* methanolic extract	Fruits	Radial Growth Method	8% (*w*/*v*)	*Phytophtohra infestans*	[50]
*S. officinalis* organic extract	Aerial part	Minimum Inhibitory Concentration Assay	12.5 mg/mL	*Candida albicans*	[46]
Crude extract (CH_2_Cl_2_–CH_3_OH) (*S. ferrugineus*)	Leaves	TLC Plates	800 mg/mL	*Candida albicans*	[48]
Egonol (*S. ferrugineus*)	Leaves	TLC Plates	5–10 μg	*Cladosporium cladosporioides*	[51]
Egonol	*S. officinalis*	Mueller Hinton Broth Method	25 μg/mL	*Candida albicans*	[49]
Homoegonol (*S. ferrugineus*)	Leaves	TLC Plates	5–10 μg	*Cladosporium cladosporioides*	[7]
Methyl β-orcinolcarboxylate (*S. suberifolius*)	Bark	Radial Growth Inhibition Assay	86.72% inhibition at 100 μg/mL	*Phomopsis cytospore*	[52]

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
