# Peer review of "Styrax* spp.: Habitat, Phenology, Phytochemicals, Biological Activity and Applications"

_plants, 2025, doi:10.3390/plants14050746_

Round 1
Reviewer 1 Report
Comments and Suggestions for Authors
Line 2, Title: Please do not use italics for "spp." (and everywhere).
Lines 18-19, Abstract: The logic is not clear. The authors write: "Many studies have been carried out on the biological applications of 18 different Styrax species, but only a few reviews have been published with a primary focus on traditional uses in folk medicine". On lines 20-22, they do not mention traditional use in folk medicine.
Lines 30-45, Introduction: Please give citation(s) for the first paragraph.
Lines 46-60, again: Please give a citation at the end of the paragraph.
Lines 62-63: use italics and normal case text in the "Styrax is an ancient Greek word meaning “fragrant resin,"
Lines 76-78: The author's statement is, "The aim of this review is to provide a comprehensive and updated analysis of the 76 scientific literature on the Styrax genus, including its classification, phenology, distribution, ..." But later they start with subchapter Distribution without classification and phenology.
Line 103: again, no citation here.
Line 138: "Yan and co-workers": give the citation number right here instead of at the end of a long sentence (like in other places).
Lines 147-149: The sentence is not clear "Styrax species display genetic variations between the wild type and commercial cultivars, since Styrax is grown for economical purposes such as extraction of benzoin resin from the bark, and genetic variations affect benzoin extraction."
General remark: Please give photos of some of the most important Styrax species.
Line 189 and 196: No citation.
Line 196: Please give a couple of samples. Which species are they?
Tables 1 and 2: Here is better to combine two very similar tables into Table 1.
Figure 2: Egonol is mentioned in several places but not presented in the Fig. 2. Please include.
Line 254: "organic extracts". Which extracts (solvents)?
Lines 280-281: ". Moreover, several studies [59] have been". If several, then please give several citations instead of one.
Line 288: Are all biological activities mentioned only in this one publication?
Line 328: ". Styrax Officinalis". Use italics and "officinalis" (small caps).
Line 375: "Phytochemicals of S. Officinalis L:" Correct as "...of S. officinalis"
Line 377: No need for "L." after the plant's Latin name.
Line 412: "(Yayla et al., 2002; Hu et al., 2019)". Use numbers instead of family names and years.
Line 420: "(E)-2-Hexenal,". Write "...hexenal".
Line 430: "geranial linalool". It must be a comma or comma with "and" between two sompounds.
Include a Discussion of the results. Also, please explain the idea described in lines 553-556 in the Discussion.
Author Response
Reviewer 1 (citations, photos, discussion of results, some clarification)
Line 2, Title: Please do not use italics for "spp." (and everywhere).
Response: The text has been modified accordingly.
Lines 18-19, Abstract: The logic is not clear. The authors write: "Many studies have been carried out on the biological applications of 18 different Styrax species, but only a few reviews have been published with a primary focus on traditional uses in folk medicine". On lines 20-22, they do not mention traditional use in folk medicine.
Response: Dear Reviewer, thank you for your suggestions and comments. The Abstract has been modified as suggested to eliminate ambiguity in the text.
Lines 30-45, Introduction: Please give citation(s) for the first paragraph.
Response: The citations were provided in the text.
Lines 46-60, again: Please give a citation at the end of the paragraph.
Response: The text has been modified accordingly.
Lines 62-63: use italics and normal case text in the "Styrax is an ancient Greek word meaning “fragrant resin,"
Response: The text has been modified accordingly.
Lines 76-78: The author's statement is, "The aim of this review is to provide a comprehensive and updated analysis of the 76 scientific literature on the Styrax genus, including its classification, phenology, distribution, ..." But later they start with subchapter Distribution without classification and phenology.
Response: The text has been modified accordingly.
Line 103: again, no citation here.
Response: Citations were included in the text.
Line 138: "Yan and co-workers": give the citation number right here instead of at the end of a long sentence (like in other places).
Response: Citations were included in the text according to the suggestion.
Lines 147-149: The sentence is not clear "Styrax species display genetic variations between the wild type and commercial cultivars, since Styrax is grown for economical purposes such as extraction of benzoin resin from the bark, and genetic variations affect benzoin extraction."
Response: Thank you for pointing out the confusion. We modified the text in the latest version
Modified text: Genetic variations are predominantly observed between the wild type and commercial cultivars of various Styrax species. Since Styrax is grown for the benzoin, these genetic variations might impact the amount and secondary metabolite contents of benzoin resins extracted from bark tissues, thereby influencing its trade value in global markets.
General remark: Please give photos of some of the most important Styrax species.
Line 189 and 196: No citation.
Response: The citation was included
Line 196: Please give a couple of samples. Which species are they?
Response: The text has been modified accordingly. We have provided the names of the species.
Tables 1 and 2: Here is better to combine two very similar tables into Table 1.
Response: We thank the reviewer for the suggestion. However, we prefer not to combine them.
Figure 2: Egonol is mentioned in several places but not presented in the Fig. 2. Please include.
Response: We included the egonol and its derivatives structures in the Fig. 5 of the manuscript.
Line 254: "organic extracts". Which extracts (solvents)?
Response: We modified the text accordingly. We included the details of the organic solvents used for the extraction in Table 1 and 2.
Lines 280-281: ". Moreover, several studies [59] have been". If several, then please give several citations instead of one.
Response: We modified the text.
Line 288: Are all biological activities mentioned only in this one publication?
Response: The reference given here (58) titled “Chemical constituents and their biological activities from genus Styrax” is a review article that provide details about the secondary metabolites and their respective biological activities. Hence, we have provided this particular reference according to the context.
Line 328: ". Styrax Officinalis". Use italics and "officinalis" (small caps).
Response: The text has been changed.
Line 375: "Phytochemicals of S. Officinalis L:" Correct as "...of S. officinalis"
Response: Done
Line 377: No need for "L." after the plant's Latin name.
Response: The text has been changed.
Line 412: "(Yayla et al., 2002; Hu et al., 2019)". Use numbers instead of family names and years.
Response: We removed the references from the text.
Line 420: "(E)-2-Hexenal,". Write "...hexenal".
Response: Done
Line 430: "geranial linalool". It must be a comma or comma with "and" between two sompounds.
Response: The text has been changed.
Include a Discussion of the results. Also, please explain the idea described in lines 553-556 in the Discussion.
Response: An explanation has been incorporated into the text.
Reviewer 2 Report
Comments and Suggestions for Authors
It is a good review of the subject. However there are some corrections to make specially about natural products chemistry:
lines 209- 212 - Some of the compounds cited are not "phenolics";
line 210 - cinnamic acid is nogt present on both resins;
line 224 - cinnamic acid is not present in siam benzoin
line 382 - There is more than one americanin. Which one do you refer?
lines 385-391 - some of the compounds are not present on figure 5 and some compounds presented on figure 5 are not benzofurans
line 391 - the compounds are not extracted from the nucleus;
lines 386-391 - structures of some of the compounds are missing on figure 5 and sometimes you should use Beta and not B. No space when the names of the compounds are cited on the text and figure 5.
line 384 - S. tokinensis or S. officinalis?
lines 392- 403 _ No space when the substituents of the compounds are cited;
lines 411-415 - Flavonoids, anthraquinones and phenols are not included in terpenoids
Author Response
It is a good review of the subject. However there are some corrections to make specially about natural products chemistry:
lines 209- 212 - Some of the compounds cited are not "phenolics";
Response: We removed the term phenolics and added general term “secondary metabolites”
line 210 - cinnamic acid is nogt present on both resins;
Response: Done; we have changed the Figure 2 legend to read as follows: “ Figure 2. Major chemical compounds found in either Sumatra benzoin and/or Siam benzoin”.
line 224 - cinnamic acid is not present in siam benzoin
Response: Done, see above response
line 382 - There is more than one americanin. Which one do you refer?
Response: In the text we mentioned as Americanin A.
lines 385-391 - some of the compounds are not present on figure 5 and some compounds presented on figure 5 are not benzofurans
Response: We have provided the structures of major compounds like egonol and its derivatives which are predominant in Styrax species and contribute for resin production and biological activities.
line 391 - the compounds are not extracted from the nucleus;
Response: We removed the term nucleus from the text.
lines 386-391 - structures of some of the compounds are missing on figure 5 and sometimes you should use Beta and not B. No space when the names of the compounds are cited on the text and figure 5.
Response: Thank you for pointing out the error. We modified the text according to your suggestions.
line 384 - S. tokinensis or S. officinalis?
Response: Styrax officinalis
lines 392- 403 _ No space when the substituents of the compounds are cited;
lines 411-415 - Flavonoids, anthraquinones and phenols are not included in terpenoids
Response: The text has been modified according to your suggestions.
Reviewer 3 Report
Comments and Suggestions for Authors
Dear Ladies and Gentlemen, Dear Journal-Team,
the interesting manuscript 'Styrax spp: Habitat, phenology, phytochemicals, biologial activity and applications' gives an overview of the Styrax plants and underlines the relevance and usage of natural compounds summarizing the characteristics, ingredients and applications of this species. It is well written. Tables and figures are sufficient.
a) Please check the number of Styrax plants in line 39.
b) Please check the headlines of the sections for uniform capital letter use.
c) Please mention the explanations of the abbreviations DPPH (diphenyl-picro-hydrazyl, line 243). MCF (Michigan cancer foundation, line 297), NOAL (no adverse effect level, line 308, and change to 'mg/kg'), EN (European norm, line 521) and TLC and EtOAc (thin layer chromatography, ethylacetate in Table 1).
d) Figure 4: Please change to 'S. officinalis' and check the spacing before the internet link.
e) Language: 1. Please check the whole manuscript for hyphen usage (Sinojackia/line 59, reclassified/line 89, subdivided, neotropical/line 92, reclassified/line 119 and 122, coworkers/line 138 and 181, intraspecific/line 152, subspecies/line 157, in depth/line 187, noncoding/line 159, midspring/line 193, antimicrobial/line 273, antiproliferative/line 291. 2. Check the writing of S. officinalis L. for uniformity (Line 375 and 473). 3. Check the spacing in Line 82 ("applications"), 321, 437, 516, 520, and the writing style in line 412. 4. What is meant with 'South to Sumatra' (line 120)? 5. Change to 'the effect varies' (Line 262) 6. Change to 'the Food and Drug Association' (line 313), and 'that the pericarp' (line 465). 7. Change to'intense distributed' (line 366). 8. Check the parentheses in line 526. 9. Check the section 9.1 Lignans and the whole manuscript for accuracy in chemical terminolgy writing.
f) References: Please check reference 6 by Perkins for accuracy, the spacing and give an English translation. Change to 'North America' in reference 12 by Gonsoulin. Check reference 23 by Wang et al., reference 45 by Icli et al., and reference 80 by Demiray et al. for the capital letter usage in the author names. Check the capital letter use in reference 48 by Pauletti et al. and reference 65 by Hamama et al. Change in reference 53 by Yanar et al. to 'In vitro'. Give an English translation for reference 68 by Amigues et al. Change to 'Styrax officinalis L. in reference 87 by Esen et al. Check reference 89 by Ertekin for the writing of the author names. Check the references 93 by Liu et al and Wang et al. for accuracy regarding the chemical terminolgy writing.
Sincerely,
Author Response
Dear Ladies and Gentlemen, Dear Journal-Team,
the interesting manuscript 'Styrax spp: Habitat, phenology, phytochemicals, biologial activity and applications' gives an overview of the Styrax plants and underlines the relevance and usage of natural compounds summarizing the characteristics, ingredients and applications of this species. It is well written. Tables and figures are sufficient.
- a) Please check the number of Styrax plants in line 39.
Response: Corrected; indeed, there are 12 genera of the Styracaceae family and not 10 gerena.
- b) Please check the headlines of the sections for uniform capital letter use.
Response: The text has been modified.
- c) Please mention the explanations of the abbreviations DPPH (diphenyl-picro-hydrazyl, line 243). MCF (Michigan cancer foundation, line 297), NOAL (no adverse effect level, line 308, and change to 'mg/kg'), EN (European norm, line 521) and TLC and EtOAc (thin layer chromatography, ethylacetate in Table 1).
Response: We expanded the terms in the text.
- d) Figure 4: Please change to 'S. officinalis' and check the spacing before the internet link.
Response: We modified the text according to your suggestion.
- e) Language: 1. Please check the whole manuscript for hyphen usage (Sinojackia/line 59, reclassified/line 89, subdivided, neotropical/line 92, reclassified/line 119 and 122, coworkers/line 138 and 181, intraspecific/line 152, subspecies/line 157, in depth/line 187, noncoding/line 159, midspring/line 193, antimicrobial/line 273, antiproliferative/line 291. 2. Check the writing of S. officinalis L. for uniformity (Line 375 and 473). 3. Check the spacing in Line 82 ("applications"), 321, 437, 516, 520, and the writing style in line 412. 4. What is meant with 'South to Sumatra' (line 120)? 5. Change to 'the effect varies' (Line 262) 6. Change to 'the Food and Drug Association' (line 313), and 'that the pericarp' (line 465). 7. Change to'intense distributed' (line 366). 8. Check the parentheses in line 526. 9. Check the section 9.1 Lignans and the whole manuscript for accuracy in chemical terminolgy writing.
Response: The manuscript has been thoroughly checked, and hyphens were removed from the suggested places.
- f) References: Please check reference 6 by Perkins for accuracy, the spacing and give an English translation.
Response: Done
Change to 'North America' in reference 12 by Gonsoulin.
Response: Done
Check reference 23 by Wang et al.,
Response: Done
reference 45 by Icli et al.,
Response: Done
and reference 80 by Demiray et al. for the capital letter usage in the author names.
Response: Done
Check the capital letter use in reference 48 by Pauletti et al.
Response: Done
and reference 65 by Hamama et al.
Response: Done
Change in reference 53 by Yanar et al. to 'In vitro'.
Response: Done
Give an English translation for reference 68 by Amigues et al.
Response: Done
Change to 'Styrax officinalis L. in reference 87 by Esen et al.
Response: Done
Check reference 89 by Ertekin for the writing of the author names.
Response: Done
Check the references 93 by Liu et al and Wang et al. for accuracy regarding the chemical terminolgy writing.
Response: Done
Round 2
Reviewer 2 Report
Comments and Suggestions for Authors
Thank you for following my suggestions to modify your paper. My opinion is that now your paper ir ready to be published